

# You can't fix what isn't broken: eight weeks of exercise do not substantially change cognitive function and biochemical markers in young and healthy adults

Joanne Gourgouvelis[1], Paul Yielder[2], Sandra T. Clarke[1], Hushyar Behbahani[2] and Bernadette Murphy[3]

[1] Department of Science, University of Ontario Institute of Technology, Oshawa, ON, Canada
[2] Department of Health Science, University of Ontario Institute of Technology, Oshawa, ON, Canada
[3] University of Ontario Institute of Technology, Oshawa, ON, Canada

## ABSTRACT

**Objective:** The benefits of exercise on brain health is well known in aging and psychiatric populations. However, the relationship between habitual exercise in young and healthy adults remains unclear. This study explored the effects an eight-week exercise prescription on cognitive function, brain-derived neurotrophic factor (BDNF) and cathepsin B (CTHB) in young and healthy adults.

**Methods:** A total of 22 low-active, young and healthy adults were recruited from a local university. A total of 12 participants performed an eight-week exercise prescription and 12 participants served as controls. Cognitive assessments, cardiorespiratory fitness and plasma BDNF and CTHB concentrations were measured at baseline and eight weeks.

**Results:** Results showed exercise improved cardiorespiratory fitness ($p = 0.044$, $d = 1.48$) with no improvements in cognitive function or no changes in plasma BDNF and CTHB concentrations.

**Conclusion:** We provide evidence that a short-term course of moderate exercise does not improve cognitive function or change plasma biochemical markers concentrations in young and healthy adults, despite mild improvements in cardiorespiratory fitness. These results suggest that cognitive health may peak during early adulthood leaving little room for improvement throughout this period of the lifespan.

Corresponding author
Bernadette Murphy,
bernadette.murphy@uoit.ca

## INTRODUCTION

Research has consistently demonstrated the health benefits of habitual exercise. Not only has exercise been shown to prevent disease, but exercise is considered an effective treatment for several medical conditions (*Naci & Ioannidis, 2013*; *Pedersen & Saltin, 2006*). More recently, considerable attention has focused on the positive effects of exercise on brain structure and function. In elderly populations, exercise has been shown to increase brain volume in selective areas such as the hippocampus and prefrontal cortex

(*Colcombe et al., 2006*; *Erickson et al., 2009*, *2011*). Additionally, exercise improves memory (*Erickson et al., 2011*; *Voss et al., 2013*), executive function (*Voss et al., 2010*), attention (*Salthouse & Davis, 2006*) and decreases cognitive processing speed (*Salthouse & Davis, 2006*). Exercise has also shown to be effective in treating mental health disorders such as anxiety (*Herring, O'Connor & Dishman, 2010*), depression (*Blumenthal et al., 1999*; *Stathopoulou et al., 2006*) and schizophrenia (*Stathopoulou et al., 2006*). Complementing human findings, the rodent literature has shown exercise to upregulate adult neurogenesis (*van Praag, Kempermann & Gage, 1999*), increase neuronal survival (*Kobilo et al., 2011*), enhance dendritic growth (*Leggio et al., 2005*), increase dendritic spine density (*Eadie, Redila & Christie, 2005*), enhance synaptic plasticity (*Farmer et al., 2004*), induce angiogenesis (*Swain et al., 2003*), enhance learning (*van Praag et al., 2005*) and improve memory (*Marlatt et al., 2012*).

Exercise activates cascades of molecular and cellular signaling mechanisms within the central nervous system. Although the precise mechanisms underlying the neurogenic effects of exercise remain unclear, a growing body of literature suggests that exercise activates neurotrophic mechanisms known to promote neuroplasticity. Most notably, brain-derived neurotrophic factor (BDNF) is emerging as a key molecule underlying the benefits of exercise on brain function (*Cotman, Berchtold & Christie, 2007*). BDNF is an activity-dependent secreted protein essential for neural growth, neural survival (*Barde, 1990*) and synaptoplastic processes critical for learning and memory (*Pang & Lu, 2004*; *Yamada, Mizuno & Nabeshima, 2002*). The brain contributes to approximately 70–80% of peripheral BDNF where it is stored and released from circulating platelets upon activation (*Fujimura et al., 2002*; *Rasmussen et al., 2009*; *Yamamoto & Gurney, 1990*). BDNF is also produced in various peripheral tissues including skeletal muscle and adipose tissue and is able to cross the blood-brain barrier (*Matthews et al., 2009*; *Pan et al., 1998*; *Sornelli et al., 2009*). In rodents, exercise rapidly increases the BDNF gene expression in brain regions involved with learning and memory formation, particularly in the hippocampus (*Berchtold et al., 2005*; *Cotman & Berchtold, 2002*; *Cotman, Berchtold & Christie, 2007*; *Neeper et al., 1995*). Similar to the increases of central BDNF expression observed in rodents, research has consistently shown that acute exercise increases peripheral BDNF concentrations in humans (*Ferris, Williams & Shen, 2007*; *Knaepen et al., 2010*; *Rojas Vega et al., 2006*). However, the literature supporting elevations in resting peripheral BDNF concentrations following a long-term exercise intervention have been mixed. In older adults, a one year moderate intensity aerobic intervention significantly increased resting plasma and serum BDNF concentrations (*Erickson et al., 2011*) that was positively associated with age (*Leckie et al., 2014*). In patients with major depressive disorder, an eight week moderate aerobic and resistance intervention significantly increased plasma BDNF concentrations (*Gourgouvelis et al., 2018*) while no change in serum BDNF concentrations were observed following a three month aerobic exercise intervention (*Krogh et al., 2014*). In young and healthy adults, a five week moderate aerobic intervention significantly increased resting plasma BDNF concentrations (*Zoladz et al., 2008*) while no increase in plasma BDNF concentrations were observed following 12 weeks of strength or 12 weeks of moderate endurance training (*Schiffer et al., 2009*) and

no change in serum BDNF concentrations following three weeks of moderate aerobic activity (*Griffin et al., 2011*). The lack of consistent findings examining the effects of long-term exercise on BDNF might be attributed to the mode of exercise, varying intensities and exercise durations between studies.

It was recently demonstrated that cathepsin B (CTHB), a cysteine proteinases produced by contracting skeletal muscle, is capable of penetrating the blood-brain barrier and upregulating both BDNF expression and hippocampal neurogenesis in wild-type mice (*Moon et al., 2016*). Following long-term running, researchers also observed an increase in plasma CTHB concentrations that was associated with improved memory performance in mice, Rhesus monkeys and humans (*Moon et al., 2016*).

Deficits in cognitive function and BDNF expression have been mainly observed in several age associated neurodegenerative diseases and psychiatric disorders (*Bocchio-Chiavetto et al., 2010*; *Diniz & Teixeira, 2011*; *Erickson & Barnes, 2003*). As such, research investigating the exercise-cognition relationship has focused on these populations, with few studies examining this relationship in young and healthy adults. The objectives of this study were to investigate the effects of a well characterized eight-week exercise intervention on cognitive function in low-active, young and healthy adults. We targeted low-active individuals as they may show a greater effect of a relatively short duration intervention. We also investigated whether changes in cognitive function were linked to changes in plasma BDNF and CTHB concentrations.

## METHODS

### Participants

A total of 22 university students (mean age = 21.10, SD = 1.27; 12 females) were recruited from a local university in Oshawa, ON, Canada. All participants completed the physical activity readiness questionnaire to screen for contraindications to exercise. Inclusion criteria included: male or female age 18–30, no history of mental health illness, low-active (exercise less than 20 min, three times weekly) and low cardiorespiratory fitness based on the Canadian Society for Exercise Physiology guidelines (*Canadian Society for Exercise Physiology, 1998*). Participants were then randomly assigned to an exercise intervention group or a control group to provide baseline and post assessment comparisons. All measures were performed at baseline and then again following the eight-week intervention (see Fig. 1 for the experiment timeline). Participants were instructed not to engage in physical activity the day of testing. This study was approved by the Ontario Institute of Technology Research Ethics Board #11979 - (10-104). All participants provided written consent.

### Neuropsychological measures

#### Cambridge neuropsychological test automated battery

Cognitive performance was evaluated using the Cambridge neuropsychological test automated battery (CANTAB) software (Cambridge Cognition, Cambridge, UK; http://www.cambridgecognition.com/cantab/cognitive-tests/). CANTAB is currently the most widely published automated neuropsychological test battery (*Wild et al., 2008*)
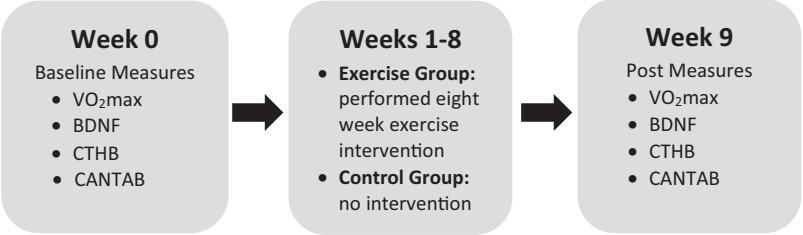

**Figure 1  Experimental timeline.**   

possessing high levels of concurrent validity and test–retest reliability (*Fowler et al., 1995*). CANTAB is an accurate, faster and more efficient method to assess cognitive functioning than traditional pen and paper tools (*Fray & Robbins, 1996*). All tests use non-verbalisable patterns and are presented on a computer touch screen in a game-like format that provides immediate feedback to reduce boredom (*Levaux et al., 2007*). The CANTAB tests included in our assessment battery assess executive function, learning and memory which have previously shown to improve following an exercise intervention (*Colcombe et al., 2004*; *Erickson et al., 2011*; *Ruscheweyh et al., 2011*; *Voss et al., 2010*) and to be sensitive to changes in the hippocampus and frontal lobes (*de Rover et al., 2011*; *Owen et al., 1991*; *Winocur et al., 2006*). A brief description of each test included in this study is provided below.

### Delayed matching to sample

This test assesses recognition memory for patterns. The subject is shown a complex visual pattern and then must choose one of four similar patterns that matches the original pattern. In some trials the sample and the choice patterns are shown simultaneously, while in others there is a delay of 0, 4 or 12 s before the choices appear. Outcome measures included accuracy and correct response latency.

### The paired associates learning

This test assesses visual memory and new learning and is sensitive to changes in the temporal and frontal lobes. In this test, boxes are displayed on the screen and are opened in a randomized order with one or more containing a pattern. Each pattern is then displayed one at a time in the middle of the screen and the participant must identify the box where the pattern was located. The participant proceeds to the next stage when all the correct locations are identified. The test has an increasing level of difficulty that ranges from two to eight patterns to be remembered. The total number of errors was used as the outcome measure for this test.

### The spatial recognition memory

This test is a measure of visual spatial recognition that uses a forced-choice discrimination paradigm in which participants must choose between previously learned and novel stimuli. The percentage of total responses correct was used as the outcome measure.

### The intra–extra dimensional set shift

This test is a measure of rule acquisition and reversal. This test assesses visual discrimination and attentional set formation, as well as maintenance, shifting and

flexibility of attention. The number of stages completed and total number of errors were used as the outcome measures.

## Plasma collection

Blood samples were collected from each participant at baseline and eight weeks by venipuncture into ethylenediaminetetraacetic acid tubes and centrifuged within 30 min. Fibrinogen containing plasma supernatant was aliquotted and stored at $-85\,°C$ until analysis. Plasma BDNF and total CTHB were quantified using enzyme-linked immunosorbant assays (ELISA) following manufacturer's protocols (R&D Systems, Minneapolis, MN, USA; BioLegend, San Diego, CA, USA). ELISA plates were read at a wavelength of 450 nm using a Synergy HTTR microplate reader (BioTek Instruments Inc., Winooski, VT, USA).

## Fitness assessment

In order to assess baseline fitness levels to determine eligibility and starting intensity for the exercise intervention baseline cardiovascular fitness was assessed with the YMCA cycle ergometer protocol recommended by the American College of Sports Medicine (*Beekley et al., 2004*; *Golding, Myers & Sinning, 1989*; *Pescatello & American College of Sports Medicine, 2014*). This protocol is a submaximal exercise that estimates maximal oxygen consumption ($VO_2$max) from heart rate (HR) measurements and perceived exertion. The protocol consisted of two or more consecutive 3-min stages at a given workload. The objective was to elevate the participant's HR to a target zone between 110 beats per minute and approximately 85% of the age-predicted maximum HR for two consecutive stages. The initial workload consisted of a 25 W workload at a cadence of 50 revolutions per minute. The workload of the subsequent stages increased by the amount specified by the YMCA protocol based on the average HR during the last 2 min of each stage. When the target HR was achieved for two consecutive stages, the test was considered complete. Participants were also assessed at eight weeks to determine changes in cardiorespiratory fitness.

## Exercise prescription

The exercise prescription was based on international guidelines of a minimum of 150 min per week of moderate to vigorous intensity aerobic exercise in combination with resistance activities two times per week, for developing and maintaining cardiorespiratory, musculoskeletal and neuromotor fitness in healthy adults (*Canadian Society for Exercise Physiology, 1998*; *Garber et al., 2011*). All exercise sessions were supervised by a qualified exercise professional. The exercise group performed one aerobic only and two resistance sessions per week on non-consecutive days for a duration of eight weeks. The exercise intensity for aerobic and resistance sessions were individualized based on each participant's target HR that ranged between 60% and 80% of their age-predicted maximum HR. Radial pulse was frequently measured throughout each aerobic and resistance exercise session to confirm that participants maintained their target HR range.

### Aerobic session

Participants were given the opportunity to choose their aerobic activity on either the treadmill, stationary bike or elliptical machine. Aerobic workloads were based on HR response and were increased by 5-min increments, over the eight weeks, reaching a maximum of 60 min per session.

### Resistance sessions

Resistance sessions combined a whole-body exercise prescription engaging the larger muscle groups. For each session, participants performed eight resistance exercises using free weights and resistance training machines. Exercises were performed consecutively in two or three supersets (with an 8–12 repetition range) to minimize rest times and to maintain target HR range. Initial workloads were individualized for each participant based on approximately 95% of their 10 repetition maximum. Subsequent workloads were increased approximately 5% once participants were able to complete three sets of 12 repetitions. Exercises were changed every four weeks to avoid adaptation; although still targeting the same muscle groups. Each session incorporated a 5 min aerobic warmup and concluded with a 15 min aerobic activity.

### Statistical analysis

All data were analyzed using Prism GraphPad software, version 6.0. Continuous data are presented as means and standard deviation (SD) and categorical data are presented as frequencies. Independent samples *t*-tests were used to compare baseline variables across groups for continuous variables and Fisher's exact tests were used to compare categorical variables. Paired *t*-tests were used to determine within-group changes from pre-post testing. A two-way analysis of variance (ANOVA) with repeated measures (Group by Time) was used to determine pre-post changes and between-group differences. Statistical significance was set at $p < 0.05$ (two-tailed) and all *p*-values were Bonferroni corrected. A modified version of Cohen's *D* ($d_{ppc2}$) specifically designed for pre-post experiments to account for any differences at baseline was used to calculate effects sizes (*Morris, 2007*).

## RESULTS

### Baseline characteristics between groups

Baseline group analyses revealed no significant differences for sex, age, BMI, BDNF and CTHB (see Table 1). One ($n = 1$) participant from the control group discontinued $VO_2$max testing due to exhaustion and was excluded from the $VO_2$max analysis. The control group showed a significantly higher mean $VO_2$max than the exercise group, $t(19) = 3.29$, $p = 0.004$, however all participants met the Poor Health Benefit Rating Zone for cardiorespiratory fitness criteria based on the Canadian Society for Exercise Physiology guidelines (*Canadian Society for Exercise Physiology, 1998*).

### Pre-post measures

A two-way ANOVA with repeated measures revealed a group by time interaction for $VO_2$max, $f(1, 19) = 7.90$, $p = 0.044$; $d = 1.48$, indicating that the exercise intervention

**Table 1 Baseline characteristics of participants.**

| Variables | Exercise group ($n = 12$) | Controls ($n = 10$) | df | p |
|---|---|---|---|---|
| Sex: (male/female) | 6/6 | 4/6 | 1 | 0.639[a] |
| Age (years) | 21.08 (1.24) | 21.16 (1.30) | 20 | 0.976[b] |
| BMI (kg/m$^2$) | 24.67 (3.68) | 22.90 (4.42) | 20 | 0.319[b] |
| VO$_2$max | 17.06 (6.12) | 26.17 (6.53)[c] | 20 | **0.004**[b] |
| BDNF (pg/ml) | 9,079 (3,480) | 8,752 (2,198) | 20 | 0.799[b] |
| CTHB (pg/ml) | 36,011 (10,778) | 42,225 (19,021) | 20 | 0.354[b] |

Notes:
Data are expressed as the mean with the standard deviation in parentheses. Significant *p*-values (< 0.05) are in bold.
BMI, body mass index; VO$_2$max, maximum oxygen consumption.
[a] Pearson's chi-square.
[b] Student's *t*-test.
[c] One missing value.

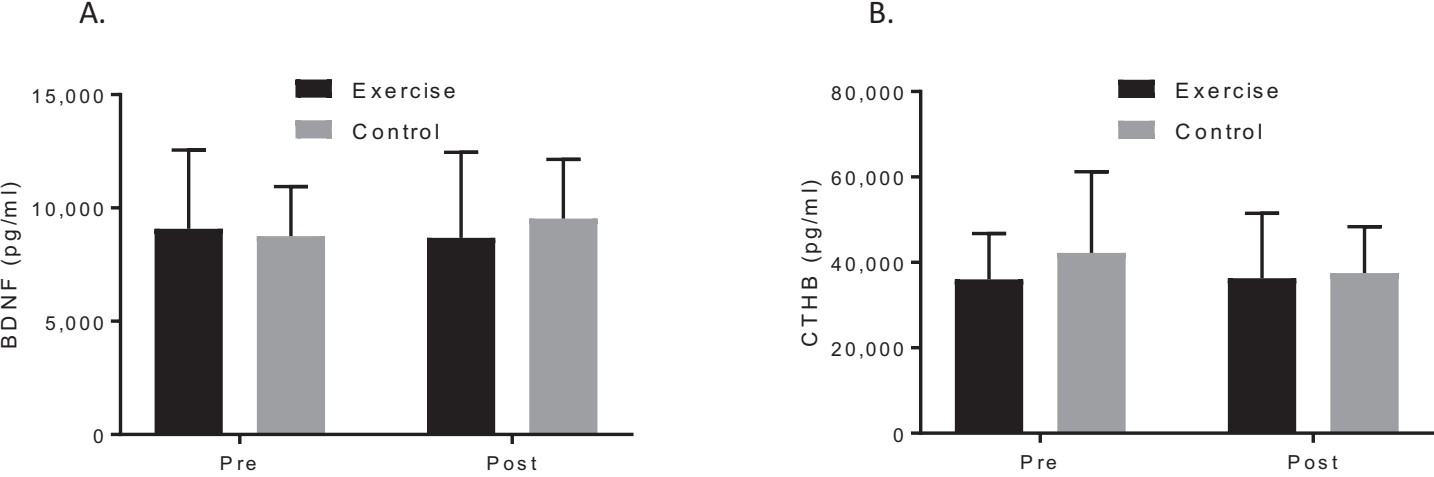

**Figure 2 Group plots illustrating pre-post biomarker changes for (A) plasma BDNF concentrations, (B) plasma CTHB concentrations.** Study sites: BDNF, brain-derived neurotropic factor; CTHB, cathepsin B.

was able to improve cardiorespiratory fitness. Biochemical marker analysis revealed no significant group-by-time effects for BDNF, $f(1, 20) = 1.29$, $p = 0.296$; $d = 0.39$, or CTHB, $f(1, 19) = 0.812$, $p = 0.379$; $d = 0.253$, see Fig. 2. Although the cognitive analyses from the CANTAB battery revealed no significant group-by-time effects for the delayed matching to sample, paired associates learning (PAL), spatial recognition memory or intra–extra dimensional set shift (see Fig. 3), a paired *t*-test revealed that the exercise group performed significantly better on the PAL task post intervention $t(11) = 3.30$, $p = 0.042$; $d = 1.35$. All pre-post results are shown in Table 2.

## DISCUSSION

This present study demonstrated that an eight-week exercise intervention, based on the minimum recommended guidelines, was able to significantly improve cardiorespiratory fitness in low-active, young and healthy adults. We provide evidence that exercise did not significantly improve cognitive function or change resting plasma BDNF or CTHB

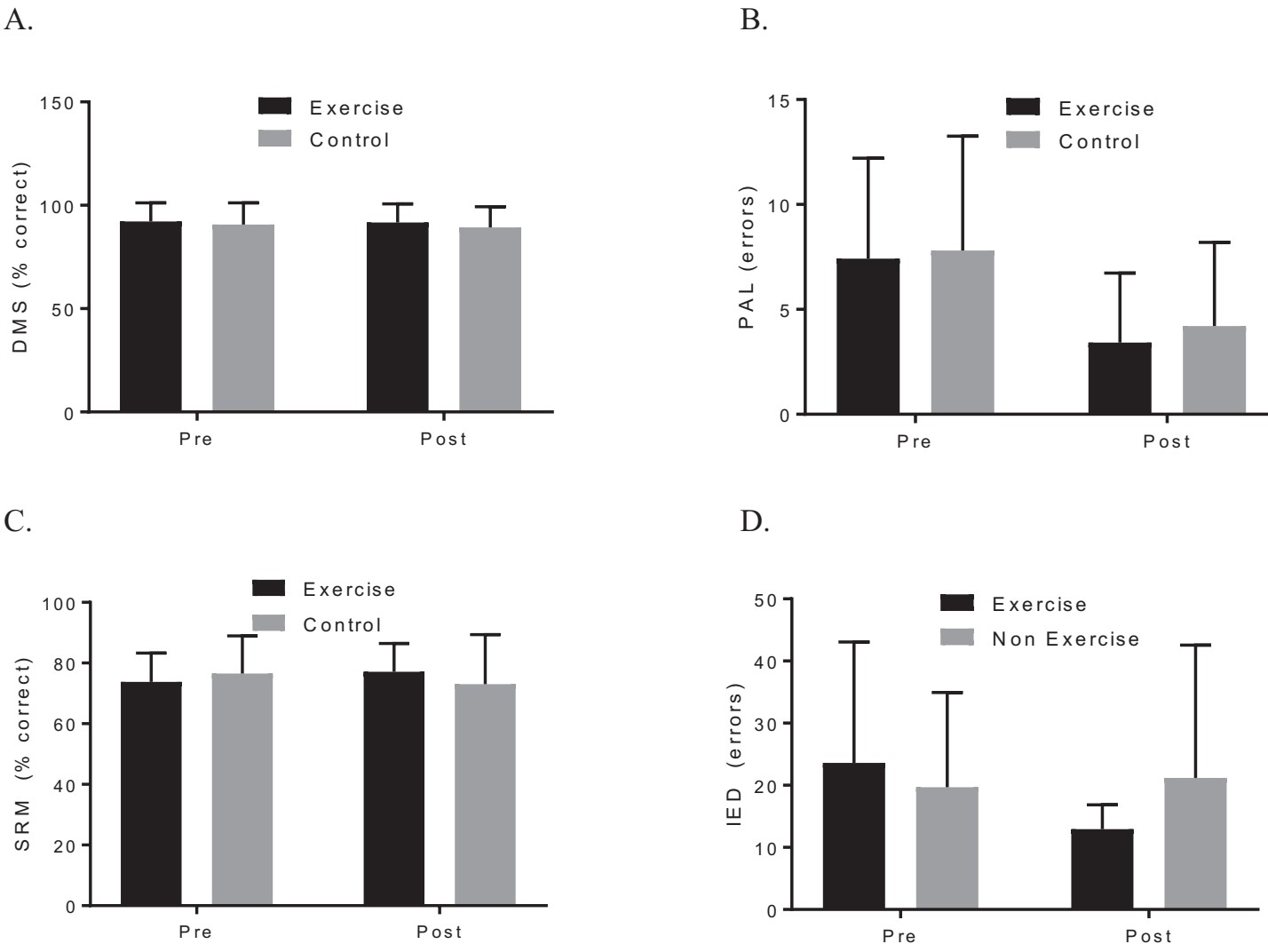

**Figure 3 Group plots illustrating pre-post CANTAB changes for (A) DMS, (B) PAL, (C) SRM, (D) IED.** Study sites: DMS, delayed matching to sample; PAL, paired associates learning; SRM, spatial recognition memory; IED, intra–extra dimensional set shift.

concentrations, despite improvements in fitness. Our findings suggest that if biomarker levels are within normal ranges at baseline, then exercise does not change these ranges in young and healthy adults. Cognitive function peaks during this time and thus to show effects of exercise on brain function may require more sensitive tests than provided by standardize cognitive testing batteries.

While the present study did not observe any significant group-by-time effects for cognitive performance, the exercise group showed improved accuracy for the PAL task. Although our findings agree with previous research (*Etnier et al., 2006*; *Felez-Nobrega et al., 2017*; *Kamijo et al., 2010*; *Themanson, Pontifex & Hillman, 2008*), there has been evidence that regular exercise may have cognitive benefits for young adults (*Aberg et al., 2009*; *Guiney & Machado, 2013*; *Themanson, Pontifex & Hillman, 2008*). The inconsistent

**Table 2 Results of pre-post changes for body mass index, fitness, BDNF, CTHB and CANTAB measures.**

| Variables | Exercise ($n = 12$) | | | | | Control ($n = 10$) | | | | | Group differences | | |
|---|---|---|---|---|---|---|---|---|---|---|---|---|---|
| | Pre mean (SD) | Post mean (SD) | $t$ | $p$ | $d$ | Pre mean (SD) | Post mean (SD) | $t$ | $p$ | $d$ | $F$ | $p$ | $d_{ppc2}$ |
| **Biological** | | | | | | | | | | | | | |
| BMI (kg/m²) | 24.66 (3.68) | 24.98 (3.19) | 0.18 | 0.86 | 0.075 | 22.90 (4.42) | 22.64 (4.35) | 0.11 | 0.92 | 0.049 | 0.040 | 0.84 | 0.14 |
| VO₂max (ml/kg/min) | 17.06 (6.12) | 25.79 (11.29) | 3.41 | **0.024** | 1.39 | 26.17 (6.53) | 25.42 (3.24) | 0.41 | 0.70[a] | 0.19 | 7.90 | **0.044**[a] | 1.48 |
| BDNF (pg/ml) | 9,079 (3,479) | 8,680 (3,781) | 0.51 | 0.62 | 0.21 | 8,751 (2,198) | 9,528 (2,612) | 1.25 | 0.24 | 0.56 | 1.29 | 0.27 | 0.39 |
| CTHB (pg/ml) | 36,011 (10,778) | 36,332 (15,178) | 0.079 | 0.939 | 0.024 | 40,972 (18,366) | 37,526 (10,826)[a] | 1.38 | 0.204 | 0.195 | 0.812 | 0.379 | 0.253 |
| **CANTAB** | | | | | | | | | | | | | |
| DMS (% correct) | 92.22 (8.91) | 91.67 (9.04) | 0.14 | 0.89 | 0.057 | 90.67 (10.52) | 89.33 (10.04) | 0.41 | 0.69 | 0.18 | 0.022 | 0.89 | 0.081 |
| DMS response latency (ms) | 3,364 (1,057) | 3,428 (863.7) | 0.143 | 0.889 | 0.058 | 3,660 (730.3) | 3,239 (494.6) | 1.91 | 0.089 | 0.85 | 0.838 | 0.37 | 0.52 |
| PAL (total errors) | 7.42 (4.80) | 3.42 (3.32) | 3.30 | **0.042** | 1.35 | 7.80 (5.453) | 4.20 (3.99) | 1.94 | 0.085 | 0.87 | 0.035 | 0.85 | 0.077 |
| SRM (% correct) | 73.75 (9.56) | 77.08 (9.41) | 1.69 | 0.12 | 0.69 | 76.50 (12.48) | 73.00 (16.36) | 0.78 | 0.45 | 0.35 | 2.20 | 0.15 | 0.61 |
| SRM response latency (ms) | 2,588 (833.3) | 2,089 (728.3) | 1.63 | 0.13 | 0.67 | 2,805 (1,160) | 2,234 (1,397) | 1.29 | 0.23 | 0.58 | 0.019 | 0.89 | 0.071 |
| IED (errors) | 23.58 (19.50)[a] | 12.92 (3.92)[a] | 1.75 | 0.108 | 0.66 | 19.70 (15.24) | 21.20 (21.35) | 0.335 | 0.745 | 0.15 | 2.41 | 0.136 | 0.68 |

**Notes:**
Data are expressed as mean with SD in parentheses. Significant $p$-values ($< 0.05$) are in bold.
All $p$-values are Bonferroni corrected.
SD, standard deviation; BMI, body mass index; VO₂max, maximal oxygen consumption; BDNF, brain-derived neurotrophic factor; CTHB, cathepsin B; CANTAB, Cambridge automated test automated battery; DMS, delayed matching to sample; PAL, paired associate learning; SRM, spatial recognition memory; IED, intra–extra dimensional set shift.
[a] One missing value.

behavioral findings in the exercise-cognition literature in young adults may be attributed to using cognitive performance measures not sensitive enough to detect small changes in cognitively high-functioning young adults. Whereas the benefits of exercise on cognitive performance observed in elderly adults may be attributed to the potentially greater gains where an age-related decline of cognitive functioning has occurred.

In an attempt to link a physiological mechanism underlying the effects of exercise on cognitive function, we measured plasma BDNF and CTHB concentrations. We did not observe any change in plasma BDNF concentrations following the exercise intervention agreeing with previous research conducted in young and healthy adults (*Griffin et al., 2011*; *Schiffer et al., 2009*). Though our study did not replicate *Moon et al. (2016)* who observed an increase in plasma CTHB following four months of aerobic exercise, our intervention was much shorter in duration. It is worth noting that the animal literature has reported reduced CTHB gene expression in the cardiac muscle of mice following five days of treadmill running (*Smuder et al., 2013*) and no change in CTHB activity in skeletal muscle following 8 h of exhaustive exercise (*Salminen, 1984*). However, little is known about the role that CTHB plays in cognitive function in humans.

## LIMITATIONS

This study was limited by the small sample size. Second, while our eight-week exercise intervention was able to significantly increase in cardiorespiratory fitness in young and healthy adults who were low-active, all participants in the exercise group remained in the poor fitness category following the intervention. As such, significant improvements in cognitive function may have been observed with an intervention of longer duration featuring additional training sessions. Future research should investigate the dose-response effects of exercise on cardiorespiratory fitness, cognitive function and biomarkers by comparing various durations of exercise. Furthermore, the CANTAB battery used for this study might not have been sensitive enough to detect changes in this high-functioning group of young and healthy adults. It is possible that the effects of exercise on cognitive behavior may only emerge when the task is extremely difficult (*Voss et al., 2011*).

## CONCLUSION

This study provides evidence that eight weeks of the minimum recommended dose of exercise does not change cognitive function, BDNF or CTHB concentrations in young and healthy, low-active adults. Our sample is representative of the a 95% of the adult population in the United States and Canada who do not meet the recommended physical activity guidelines for health benefits (*Colley et al., 2011*; *Troiano et al., 2008*). In order to understand the magnitude of the effects of exercise on cognitive function, and the relationship between cardiorespiratory fitness and cognitive function across the life span, it is critical that we understand the impact of exercise on cognitive function in young and healthy populations that are physically active and inactive.

## ACKNOWLEDGEMENTS

The authors would like to thank Joanne Free and Julia Green-Johnson from the University of Ontario of Technology, for their assistance, technical expertise and guidance.

### Funding

Joanne Gourgouvelis received the Ontario Graduate Scholarship. The funders had no role in study design, data collection and analysis, decision to publish, or preparation of the manuscript.

### Grant Disclosures

The following grant information was disclosed by the authors:
Ontario Graduate Scholarship.

### Competing Interests

The authors declare that they have no competing interests.

### Author Contributions

- Joanne Gourgouvelis conceived and designed the experiments, performed the experiments, analyzed the data, contributed reagents/materials/analysis tools, prepared figures and/or tables, authored or reviewed drafts of the paper, approved the final draft.
- Paul Yielder conceived and designed the experiments, analyzed the data, authored or reviewed drafts of the paper.
- Sandra T. Clarke performed the experiments, analyzed the data, contributed reagents/materials/analysis tools, authored or reviewed drafts of the paper.
- Hushyar Behbahani performed the experiments, analyzed the data, contributed reagents/materials/analysis tools, authored or reviewed drafts of the paper.
- Bernadette Murphy conceived and designed the experiments, analyzed the data, prepared figures and/or tables, authored or reviewed drafts of the paper, approved the final draft.

### Human Ethics

The following information was supplied relating to ethical approvals (i.e., approving body and any reference numbers):

This study was approved by the Ontario Institute of Technology Research Ethics Board #11979 - (10–104).

### Data Availability

The raw data are provided in the Supplemental Dataset Files.

### Supplemental Information

Supplemental information for this article can be found online at http://dx.doi.org/10.7717/peerj.4675#supplemental-information.

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
