# Peer review of "You can’t fix what isn’t broken: eight weeks of exercise do not substantially change cognitive function and biochemical markers in young and healthy adults"

_PeerJ, doi:10.7717/peerj.4675_

## Round 0.1 · original submission · Major Revisions

I now have received two reviewers' comments. Although both reviewers expressed their interest in your study, several aspects of this manuscript should be revised to improve its clarity. Their observations are presented with clarity so I'll not risk confusing matters by belaboring or reiterating their comments. While I might quibble with the occasional point, I note that I regard the reviewers' opinions as substantive and well-informed. I believe that all of the highlighted reservations require contemplation and appropriate attention in revising the document if it is to contribute appropriately to Peerj and the extant literature. Please revise or refute according to the two reviewers' comments and provide a point by point reply in addition to the revised manuscript.

Tsung-Min Hung, Ph.D.
PeerJ editor
Distinguished professor
Department of Physical Education
National Taiwan Normal University

Reviewer 1 ·

Basic reporting

See below

Experimental design

See below

Validity of the findings

See below

Additional comments

The manuscript presents an interesting approach to investigating the neural mechanisms of exercise-induced neural plasticity, a topic of emerging interest in neuroscience. The paper is very well written, the methods and rationale are clearly explained. I have some suggestions related to the statistical approach that may make the finally conclusions easier to accept.
Considerations
1. Line 75-76. Expand on the other sources of BDNF available.
2. 105. References need after this statement ending in humans.
3. A figure showing the timeline of data acquisition and intervention details for both groups would benefit the reader.
4. Results. The text is heavily laden with F stats. Perhaps this section would be easier to read if these are inserted into a table rather than the text.
5. Results. As can be seen from Table 1, the controls and exercise group were not matched for fitness level. As a result, the statistical analysis may be better served by performing separate ANOVA’s on each group independently. As it stands, the improvement in the exercise group leads them to have fitness levels that match the controls. Therefore, the controls aren’t really ‘controls’ for fitness level. They are controls for the type of intervention. Their data can be analyzed independent of the exercise group and this is unlikely to impact the main conclusion of this paper.
6. Results. Table 1. Since the n=12 and 10, the line indicating Sex (male/female) should have values that sum to 12 and 10 respectively. At present, these sum to 18 and 14.
7. Title and abstract. The title and abstract should better articulate the finding that exercise of this duration does not lead to substantial improvements in fitness and changes in cognition and neurotrophins. It is true that the small improvement in fitness level has not changed the overall fitness level of the exercise group and this should be emphasized in the abstract and the title.

Reviewer 2 ·

Basic reporting

This manuscript examined the effect of eight weeks of intervention combining aerobic and resistance exercise on several aspects of cognition assessed by CANTAB and biomarkers including resting BDNF and CTHB in young and healthy, but inactive and low-fit adults. The results show no changes in cognitive performance and biomarkers despite improved cardiorespiratory fitness. While respecting the authors’ work on this study, the lack of discussion regarding how the research question was formed, how this study was designed, and how the findings should be interpreted makes this manuscript unacceptable for publication. The null finding on the cognitive and biomarker outcomes is fine but the authors really need to do a better job at writing for better clarity. Thus, a major revision is necessary before this manuscript being considered for publication.


 Line 97: After the literature review, the authors concluded that inconsistent BDNF findings may be due to the heterogeneity of exercise intervention. Why and how? Please explain. Did all previous studies have no baseline PA measures? What was the consequence of not having baseline PA?

 Line 109: It is not clear why this study was necessary. The authors cited a few prior studies in young adult but how the existing evidence inspired this study? Overall the introduction does a good job in reviewing the literature but does not really explain what knowledge gap this study is going to fill. Perhaps the addition of CTHB?

 Line 196: Not sure what does this sentence “exercise group performed one aerobic only and two sessions per week ….of eight weeks” means.

 Line 237: The authors should provide stats for the VO2max analysis. Was that a significant Group by Time interaction? How this interaction was decomposed? What post-hoc test was used? Was a corrected p-value used? Please clarify both in the method and results sections.

 Line 241: Are these stats for response accuracy since the following sentence is for correct response latencies?

 Line 262-270: Does this entire paragraph solely base on Etnier et al. (2006)? It seems like some more citations are needed in this paragraph.

 Line 267: Maybe I miss something here. But if not, I don’t know where is the statement “fitness was negatively predictive of cognitive performance for children” coming from… There are a lot of studies showing positive associations between childhood fitness and performance of various aspects of cognition (Hillman et al., 2008; Khan & Hillman, 2014). This is a completely false statement basing on no citation.

 Line 274-297: Not sure why this paragraph is included. We know the literature, but this does not help explain your findings. Perhaps the purpose of this paragraph was to suggest that covert measures of cognition such as ERP and brain activation are needed to observe the intervention effect on cognition. The authors need to explain better here.

 Line 307: Did you mean little is known in human?

 Line 316: Various duration of exercise session or duration of intervention?

 Line 323: Untrained is not equal to inactive and low-fit. Please define the participants well in the manuscript.

 Line 326: I can’t follow. How come your sample being categorized as sedentary can lead to a conclusion that neurodegenerative changes observed in aging and psychiatric populations are likely confounded by sedentary behavior and poor fitness?

 Line 329: “in” young and healthy population?

Experimental design

 Why targeted low-active and low-fit individuals? What’s the rationale for targeting this population? The authors may have to address this in the introduction. What is the rationale for N = 22? Did you conduct a power analysis?

 Line 149: What does “or there is a brief delay of 0, 4, and 12 seconds between these steps” mean?

 Where did the cognitive assessment take place? Were participants told to not engage in any physical activity prior to the pre-test and post-test sessions since exercise may have acute effects on cognitive performance? More description regarding the testing days is needed.

Validity of the findings

 How about the control group? What did they do during the intervention period?

 The authors should report HR during the intervention to demonstrate that exercise intervention was successfully administered.

 Does change of fitness correlate with any change of dependent measures?

---

## Round 0.2 · accepted · Accept

I have now received reviewers’ comment and were satisfied with your reply and revisions from previous comments. You and your coauthors have my congratulations. Thank you for choosing PeerJ as a venue for publishing your research work and I look forward to receiving more of your work in the future.

Tsung-Min Hung, Ph.D.
PeerJ editor
Distinguished professor
Department of Physical Education
National Taiwan Normal University

# Reviewer 2 ·

Basic reporting

no comment

Experimental design

no comment

Validity of the findings

no comment

Additional comments

no comment